# Paying attention to cardiac surgical risk: An interpretable machine learning approach using an uncertainty-aware attentive neural network

Jahan C. Penny-Dimri[1]*, Christoph Bergmeir[2,3], Christopher M. Reid[4], Jenni Williams-Spence[4], Andrew D. Cochrane[1], Julian A. Smith[1]

1 Department of Surgery, School of Clinical Sciences at Monash Health, Monash University, Melbourne, Vic, Australia, 2 Department of Computer Science and Artificial Intelligence, University of Granada, Granada, Spain, 3 Department of Data Science and Artificial Intelligence, Faculty of Information Technology, Monash University, Melbourne, Vic, Australia, 4 Department of Epidemiology and Preventive Medicine, Monash University, Melbourne, Vic, Australia

* jahan.penny-dimri@monash.edu

**Data Availability Statement:** All relevant data are within the paper and its Supporting Information files.

## Abstract

Machine learning (ML) is increasingly applied to predict adverse postoperative outcomes in cardiac surgery. Commonly used ML models fail to translate to clinical practice due to absent model explainability, limited uncertainty quantification, and no flexibility to missing data. We aimed to develop and benchmark a novel ML approach, the uncertainty-aware attention network (UAN), to overcome these common limitations. Two Bayesian uncertainty quantification methods were tested, generalized variational inference (GVI) or a posterior network (PN). The UAN models were compared with an ensemble of XGBoost models and a Bayesian logistic regression model (LR) with imputation. The derivation datasets consisted of 153,932 surgery events from the Australian and New Zealand Society of Cardiac and Thoracic Surgeons (ANZSCTS) Cardiac Surgery Database. An external validation consisted of 7343 surgery events which were extracted from the Medical Information Mart for Intensive Care (MIMIC) III critical care dataset. The highest performing model on the external validation dataset was a UAN-GVI with an area under the receiver operating characteristic curve (AUC) of 0.78 (0.01). Model performance improved on high confidence samples with an AUC of 0.81 (0.01). Confidence calibration for aleatoric uncertainty was excellent for all models. Calibration for epistemic uncertainty was more variable, with an ensemble of XGBoost models performing the best with an AUC of 0.84 (0.08). Epistemic uncertainty was improved using the PN approach, compared to GVI. UAN is able to use an interpretable and flexible deep learning approach to provide estimates of model uncertainty alongside state-of-the-art predictions. The model has been made freely available as an easy-to-use web application demonstrating that by designing uncertainty-aware models with innately explainable predictions deep learning may become more suitable for routine clinical use.

**Funding:** The ANZSCTS Cardiac Surgery Database Program is funded by the Department of Health (Victoria), the Clinical Excellence Commission (NSW), Queensland Health (QLD), and funding from individual cardiac surgical units participating in the registry. ANZSCTS Database Research activities are supported through a National Health and Medical Research Council Principal Research Fellowship (APP 1136372) and Program Grant (APP 1092642) awarded to C.M. Reid. This particular study received no additional funding or financial support. The funders had no role in study design, data collection and analysis, decision to publish, or preparation of the manuscript.

**Competing interests:** The authors have declared that no competing interests exist.

**Abbreviations:** ANZSCTS, Australian and New Zealand Society of Cardiac and Thoracic Surgeons; AUC, Area under the receiver operating characteristics curve; EDT, Ensemble decision tree; eGFR, Estimated glomerular filtration rate; GVI, Generalized variational inference; LR, Logistic regression; MIMIC, Medical Information Mart for Intensive Care; ML, Machine Learning; PN, Posterior network; t-SNE, t-distributed stochastic neighbor embedding; UAN, Uncertainty-aware attention network.

# Introduction

Machine learning (ML) is increasingly being applied to risk stratification and prediction of postoperative outcomes in cardiac surgery [1]. The extreme physiological demands of cardiac surgery make the development of effective risk stratification tools an important strategy for improving patient care [2]. Currently, the most widely used tools are clinical scores, which are derived from logistic regression (LR) models [3–5]. Modern approaches, such as ensemble decision tree (EDT) models or deep neural networks, have had limited success improving upon the performance or the interpretability of the standard linear regression methods [6–9].

A ML model must demonstrate four key qualities to be safely applied in a healthcare setting. It needs to be performant, interpretable, uncertainty-aware, and robust to incomplete data [10]. The current gold standard in risk stratification for cardiac surgery is the LR model, which has proven performant, interpretable, and to an extent uncertainty aware [11]. EDT models, such as gradient boosting machines, have also been shown to be highly performant, flexible to missing data, and globally interpretable [12]. While these models each fulfill some of the qualities needed, their limitations restrict widespread adoption into clinical practice.

## Aims and hypothesis

Our aims were: 1. Develop a novel ML model, the uncertainty-aware attention network (UAN), that fulfilled the four key qualities of an ML model. 2. Benchmark predictive performance against two current gold standard models. These models were a popular EDT, XGBoost, and a Bayesian LR model. 3. Benchmark uncertainty quantification against the gold standard models. 4. Benchmark performance with missingess against common imputation methods. 5. Demonstrate the predictive explanations generated from the UAN.

We hypothesized that the UAN would be as performant as the benchmark models, quantify uncertainty well, and provide individual explanations for each prediction, visualized with a heat map.

# Methods

## Study population

The Australian and New Zealand Society of Cardiac and Thoracic Surgeons (ANZSCTS) Cardiac Surgery Database registry recorded 153,932 cardiac surgery events in 151,078 unique patients from June 2001 to December 2019, captured at 32 centers in Australia. As the Database stores sensitive patient information, it is not publicly available. Criteria for inclusion within the database was any patient undergoing cardiac surgery, other thoracic surgery using cardiopulmonary bypass (CPB), or pericardiectomy for constrictive pericarditis, where performed on or off CPB [13].

The third version of the Medical Information Mart for Intensive Care (MIMIC III), a large single-center database, contained data from 53,423 distinct hospital admissions for 38,597 distinct adult patients admitted to the intensive care unit at Beth Israel Deaconess Medical Center in Boston, Massachusetts between 2001 and 2012 [14]. A surgical cohort undergoing either coronary artery bypass or valvular surgery was subsetted and used for external validation of models fitted to the ANZSCTS database.

## Variable selection

41 preoperative, intraoperative, and early postoperative variables available across either dataset were analyzed for inclusion in the predictive modeling. Included variables and their definitions are listed in S1 of the supplemental materials.

## Outcomes

Eight clinically significant outcomes were modeled, however, only 30 day mortality was available in both datasets for external validation. Outcomes and their definitions are listed in S2 of the supplemental materials. While multiple clinically important outcomes were tested, for simplicity only the results for 30 day mortality on the external validation dataset will be reported in the rest of the manuscript. Performance and calibration results for all outcomes on training and validation datasets are available in the S4 and S5 of the supplemental materials.

## Data preprocessing

Three different processing strategies were utilized. Firstly, a minimal processing strategy was used for models capable of handling missing data, whereby all patient data were included in training and validation. Secondly, a partial dataset was further subsetted, including only variables present in both external and derivation datasets and excluding patient data with missing values. The partial dataset was used to train and validate models incapable of handling missing data. Thirdly, two imputation strategies were used to replace missing values and train the baseline models and compare performance against the UAN without imputation. Simple imputation using mode and mean replacement, and multivariate imputation with chained equations (MICE) were used [15]. Throughout this process, outlier values for each variable were relabelled as missing. The data processing pipeline for the datasets described above is visualizes in Fig 1. Definitions for outliers can be found in the S1 of the supplemental materials.

## Machine learning models

**Uncertainty-aware attention network.** At the core of the UAN, is the attention function. The attention function is a method of mapping outcome vectors, called 'queries,' to inputs that are transformed with neural networks to 'keys' and values' [16]. The queries, keys, and values are combined using dot product operations, allowing for processing of sequences of different lengths and masking of specific values [16]. Dot-product attention has been very successful in processing sequences, but can be extended to processing incomplete inputs [16].

In predictive tasks with missing values, the question being answered is 'what is the probability of the outcome given the observed values?' An attention model that learns a posterior distribution with Bayesian methods can effectively answer that question for multiple outcomes, in a principled and consistent way, without imputation, excluding data, or limiting the scope of input variables to the model. The attention function is also innately interpretable, whereby the attention the model is paying to each variable can be visualized in a heat-map, allowing for predictive explainability.

Two different approaches to modeling uncertainty were applied to the UAN. The first was to learn the parameters of a posterior beta distribution using generalized variational inference (GVI) [17]. The second was to learn the parameters of a posterior beta distribution with the addition of posterior networks (PN), which have the reported advantage of better uncertainty on out-of-distribution inputs [18].

A further description and implementation details of the UAN can be found in S1 Appendix. An example code-base for the model and web application for hosting the model can be found at https://github.com/jahanpd/UAN.

**Bayesian logistic regression.** A Bayesian logistic regression model is a generalized linear model which learns a distribution over the coefficients in the model [11]. It was trained using variational inference [19].

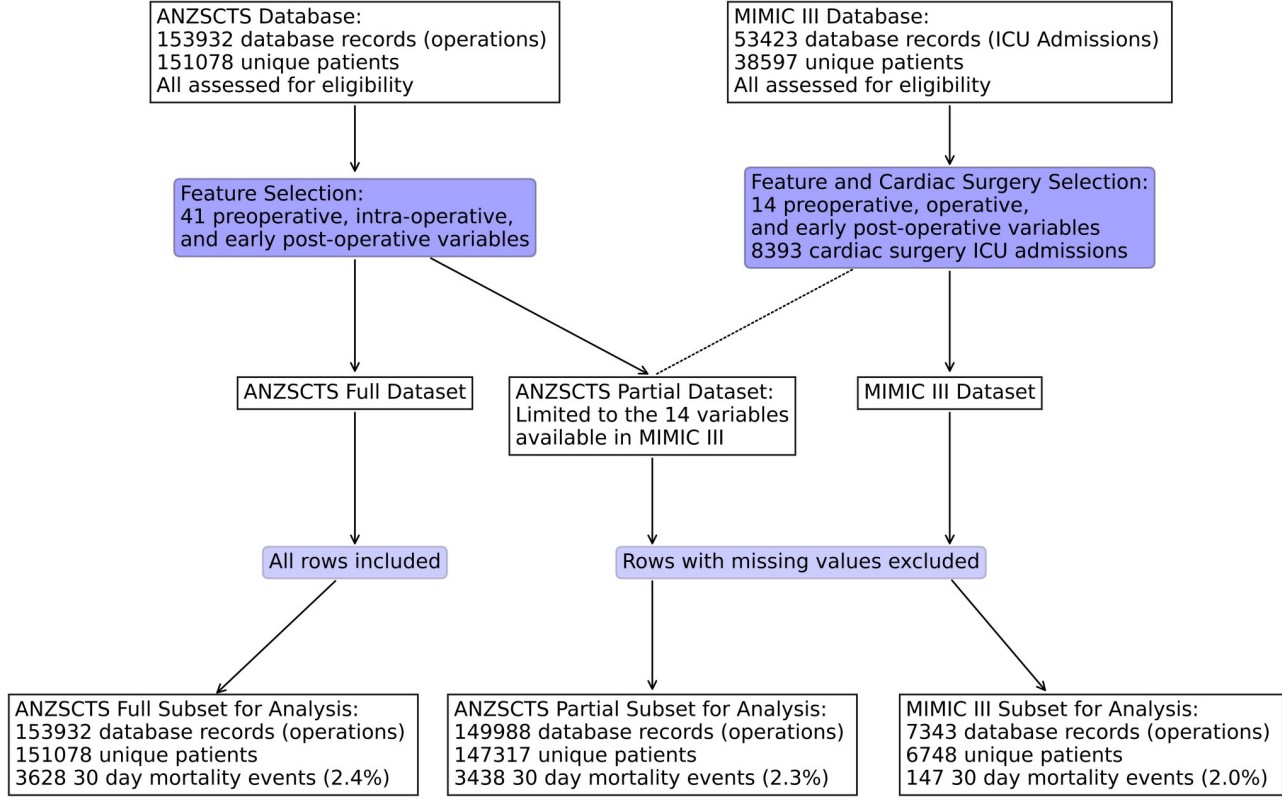

**Fig 1. Flow diagram depicting included and excluded data following cleaning and subsetting for the full and partial ANZSCTS datasets, and the MIMIC III dataset.**

**Extreme gradient boosting (XGBoost).** XGBoost is a popular gradient boosting decision tree model [12]. In order to estimate uncertainty, an ensemble of XGBoost models was trained on bootstrapped sampling of the training dataset [20].

**Implementations.** All models were implemented using open source software, with the UAN and LR models implemented in Pytorch, and XGBoost from the publicly available python package [12, 21]. Imputation was performed using the implementations in the python package scikit-learn [22].

## Training, visualization, and statistical methodology

Models were trained on rebalanced datasets using randomly combined majority and minority class undersampling and oversampling [23]. Model performance was assessed using 5-fold cross-validation repeated 4 times. Imputation, rebalancing, and standardization of data were all calibrated using the training set on a per-fold basis. Performance metrics were the area under the receiver operating characteristic curve (AUC), sensitivity, and specificity. Pairwise testing of distributional differences was performed using a student T-test.

To assess confidence calibration we distinguished between two forms of uncertainty, which were aleatoric and epistemic. Aleatoric uncertainty arises from irreducible noise intrinsic to the data, and epistemic uncertainty from absent knowledge due to unseen data [18]. Confidence calibration was assessed using the AUC with labels of 1 for correct predictions and 0 for incorrect. Aleatoric confidence calibration was determined using the probability of the predicted class as scores. Epistemic confidence calibration was determined using either the

maximum parameter of the beta distribution for the UAN, or the inverse empirical variance for the LR model or ensemble models, as scores. The Brier score was measured to further assess aleatoric uncertainty calibration for each model. Performance of the model was also measured on high confidence samples, which was defined as a prediction where the 95% credible or confidence interval did not overlap with the baseline risk.

Model interpretability for UAN predictions was visualized using an attention heatmap depicting the outcomes and feature importance from the partial dataset. The UAN model was further interrogated for model fit by plotting methods in two ways. Firstly, the univariate predicted probability for mortality was plotted against the input range of four continuous features, which were age, body mass index, estimated glomerular filtration rate (eGFR), and preoperative creatinine. This is possible as the UAN is flexible to input features and able to provide predictions with only one input variable, in contrast to the other models. Secondly, the embedded representations of the input features and outcomes were visualized using t-distributed stochastic neighbor embedding (t-SNE) to 2-dimensional space. This plot shows how the UAN learns how closely related features or outcomes are to each other as demonstrated by the proximity in the plot.

## Ethics

This project was conducted in concordance with the National Health and Medical Research Council (NHMRC) National Statement on Ethical Conduct in Human Research, with approval from the Monash University Human Research Ethics Committee (HREC) approval number 2020-24850-45439. The ANZSCTS National Database has approval for data collection from Monash University and Alfred Health HREC utilizing an opt-out system for patient consent with the data policy and patient information sheet available in the supplemental materials.

## Results

### Data subsets

All 153,932 cardiac surgical procedures in 151,078 unique patients were included from the ANZSCTS Database, and 7,343 cardiac surgical procedures in 6,748 unique patients were subsetted from the MIMIC III database. 149,988 procedures in 147,317 unique patients were further subsetted from the ANZSCTS Database for the partial dataset. Patient characteristics for the partial dataset and the MIMIC III dataset are presented in Table 1, and for the full dataset in S3 of the supplemental materials.

As expected, there were many differences detected between the derivation and external validation datasets. The largest were in a recorded history of cardiac arrhythmia with 17. 17% and 52.34%, and a history of smoking with 57.35% and 15.92% for the ANZSCTS and MIMIC III datasets respectively.

### Model performance and uncertainty calibration

For the main outcome of AUC, the highest performing model was the UAN-GVI and UAN-PN with scores of 0.78 (0.01) (Table 2). All model performance improved when measured on high confidence samples, with the UAN-GVI and UAN-PN performing the best with scores of 0.82 (0.01) (Table 3).

Confidence calibration for aleatoric uncertainty was excellent for all models, achieving a score of close to, or exactly 1.0. Calibration for epistemic uncertainty was more variable, with the ensemble of XGBoost models with MICE performing the best with a score of 0.84 (0.08) (Table 3). The UAN-GVI and LR with simple imputation had the worst epistemic calibration

**Table 1. Patient characteristics of the partial database subsets.**

| Variable | ANZSCTS Subset Mean (SD) or % | MIMIC III Subset Mean (SD) or % |
|---|---|---|
| Age (years) | 65.61 (12.89) | 69.57 (27.69) |
| Sex | 73.20% | 68.19% |
| Body Mass Index | 28.48 (5.43) | 28.59 (5.77) |
| Insurance | | |
| Private | 25.64% | 36.63% |
| DVA | 1.34% | 0% |
| Medicare | 71.74% | 56.50% |
| Self Insured | 0.31% | 0.39% |
| Overseas | 0.52% | 0% |
| Other | 0.44% | 6.47% |
| History of Arrhythmia | 17.17% | 52.34% |
| History of Smoking | 57.35% | 15.92% |
| History of Diabetes | 29.19% | 33.26% |
| History of Hypercholesterolaemia | 66.08% | 55.07% |
| History of Hypertension | 72.08% | 70.26% |
| History of Peripheral Vascular Disease | 8.93% | 14.88% |
| History of Heart Failure | 20.99% | 29.93% |
| Type of Operation | | |
| CABG alone | 53.46% | 58.18% |
| Isolated Valve | 20.47% | 26.27% |
| CABG+Valve | 10.20% | 15.55% |
| Other | 15.87% | 0% |
| Preoperative Creatinine (micromol/L) | 101.77 (85.71) | 109.65 (126.57) |
| Hours in ICU | 83.80 (147.93) | 92.67 (129.66) |
| 30 Day Mortality | 3438 (2.29) | 147 (2.00) |

with scores of 0.63 (0.10) and 0.57 (0.14), respectively. Compared to the UAN-GVI, the PN approach had better epistemic calibration with a score of 0.63 (0.10). The Brier score was best in the XGBoost and worst in the LR model with scores of 0.12 (0.07) and 0.23 (0.01) respectively.

Statistical testing is reported in S5–S12 Tables of the supplementary materials.

## Interpretability

The attention paid to each feature by the UAN-PN model was visualized for high risk, low risk and uncertain risk prediction (Fig 2).

**Table 2. Model performance on the MIMIC III dataset for 30 day mortality.**

| Model | AUC | Sensitivity | Specificity |
|---|---|---|---|
| LR | 0.75 (0.00) | 0.72 (0.02) | 0.67 (0.01) |
| LR (simple imputation) | 0.66 (0.01) | 0.21 (0.03) | **0.90 (0.01)** |
| LR (MICE) | 0.73 (0.00) | 0.53 (0.02) | 0.78 (0.01) |
| XGBoost | **0.78 (0.02)** | 0.47 (0.29) | 0.84 (0.12) |
| XGBoost (simple imputation) | 0.57 (0.03) | 0.36 (0.22) | 0.70 (0.19) |
| XGBoost (MICE) | 0.75 (0.01) | 0.63 (0.09) | 0.72 (0.07) |
| UAN (GVI) | **0.78 (0.01)** | **0.76 (0.05)** | 0.66 (0.05) |
| UAN (PN) | **0.78 (0.01)** | **0.76 (0.05)** | 0.66 (0.05) |

**Table 3. Uncertainty calibration on the MIMIC III dataset for 30 day mortality.**

| Model | AUC High Confidence | AUC Aleatoric | AUC Epistemic | Brier Score |
|---|---|---|---|---|
| LR | 0.76 (0.00) | **1.00 (0.00)** | 0.69 (0.01) | 0.23 (0.01) |
| LR (simple imputation) | 0.67 (0.01) | 0.98 (0.00) | 0.60 (0.01) | 0.13 (0.00) |
| LR (MICE) | 0.75 (0.00) | 0.99 (0.00) | 0.70 (0.01) | 0.17 (0.00) |
| XGBoost | 0.80 (0.06) | **0.99 (0.01)** | 0.79 (0.15) | **0.12 (0.07)** |
| XGBoost (simple imputation) | 0.54 (0.11) | 0.99 (0.01) | 0.59 (0.13) | 0.20 (0.07) |
| XGBoost (MICE) | 0.79 (0.01) | **1.00 (0.00)** | **0.88 (0.05)** | 0.17 (0.04) |
| UAN (GVI) | **0.82 (0.01)** | **1.00 (0.00)** | 0.57 (0.14) | 0.22 (0.02) |
| UAN (PN) | **0.82 (0.01)** | **1.00 (0.00)** | 0.64 (0.10) | 0.22 (0.02) |

## Model interrogation

The univariate predicted probability distribution for age, body mass index, eGFR, and preoperative creatinine demonstrated sensible trends. The probability of 30-day mortality increased with age, body mass index, and preoperative creatinine and decreased with eGFR (Fig 3).

The cluster-mapped features showed that the UAN learned feature embeddings that were similar for features that were related or had similar effects on the predicted outcome (Fig 4).

## Web application

The UAN-PN was made available for test use at www.cardiac-ml.com.

## Discussion

This is the first description of an uncertainty-aware attention-based neural network. The results presented demonstrate the UAN to have performance as good or better than traditional benchmarks, as well as acceptable uncertainty calibration. The success of ML across many industries has not translated to healthcare, as performance alone does not overcome clinicians' lack of trust in black-box model predictions [24]. Key to developing clinician trust and use is flexibility to incomplete data, interpretable predictions and uncertainty assessment. We have demonstrated that the UAN provides all of these features.

### Incomplete data

Healthcare data is often incomplete, which presents a problem for many models. LR, and their derivative scoring tools, are only valid when the clinician has all variables at hand [11]. Excluding patient data with missing values or performing imputation is likely to introduce bias [25]. The UAN overcomes this problem by applying the attention function in a novel approach allowing extremely flexible inference with 1 or more features available. In this work, we demonstrated this capability by training the model on 41 features, however, the external validation dataset only had 14 features from which to generate predictions. Importantly, in this setting, the UAN is as good or better than imputing data before inference or training.

### Model interpretability

For clinicians to trust a predictive model, the decision-making mechanism must be transparent. In LR, coefficients associated with each variable are interpretable as the importance the fitted model ascribes to that variable [11]. Similarly, for EDT, methods exist to interpret the importance the model weights for each feature. These are, however, global explanations and do not provide transparency for a specific prediction [12]. Further explanatory methods can

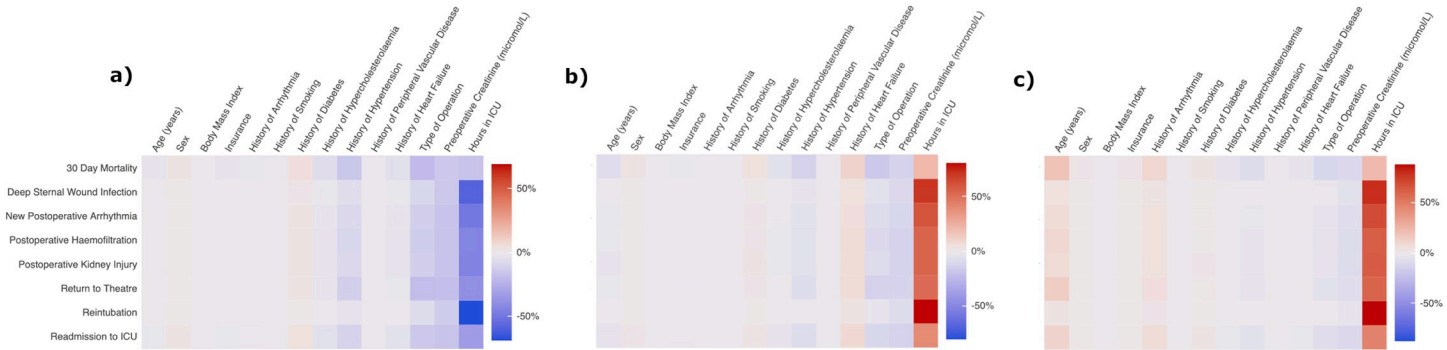

**Fig 2. Attention heat maps for 3 samples from the MIMIC III dataset with high, low and baseline population risk of a postoperative complication.** These plots provide a visual explanation for how the model is weighting each feature with respect to each of the outcomes. The patient in a) had a low probability of 30 day mortality of 0.10, and their characteristics were age 51, male, BMI 25.4, privately insured, no history of arrhythmia, diabetes, smoking, high cholesterol, hypertension, peripheral vascular disease or heart failure, an isolated CABG, preoperative creatinine of 53 and ICU length of stay of 43 hours. The patient in b) had a baseline probability of 30 day mortality of 0.5, and their characteristics were age 61, male, BMI 21.0, privately insured, no history of arrhythmia, smoking, diabetes, high cholesterol, hypertension, peripheral vascular disease, a positive history for heart failure, an isolated CABG, preoperative creatinine of 61.9 and ICU length of stay of 146 hours. The patient in c) had a high probability of 30 day mortality of 0.85, and their characteristics were age 86, male, BMI 23.0, insured under medicare, no history of smoking, diabetes, high cholesterol, hypertension, peripheral vascular disease, or heart disease, a positive history for cardiac arrhythmia, a combined CABG and valve operation, a preoperative creatinine of 106 and ICU length of stay of 317 hours.

be applied to trained models in order to provide individualized explanations [26]. This approach is limited by needing to optimize another model based on the output of the trained model, which can be inefficient or result in inaccurate explanations [26–28]. The UAN has inherent interpretability by visualizing the attention paid to each variable, allowing for individualized prediction level explanations (Fig 2). In this way, the UAN provides individualized risk

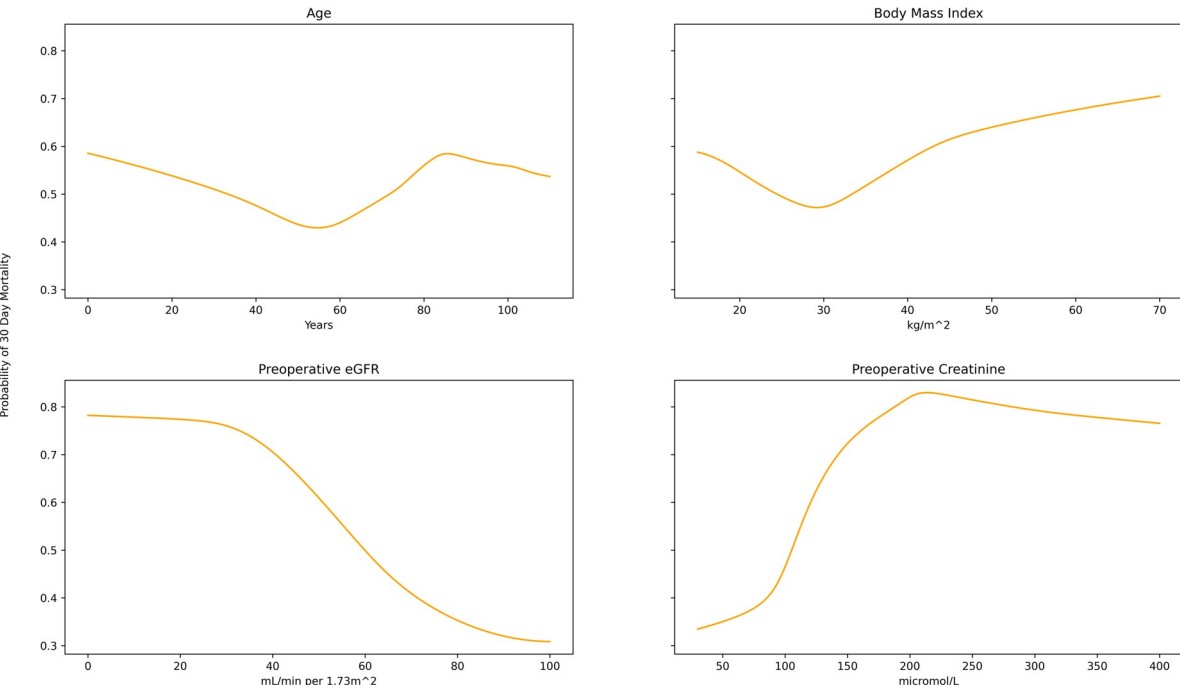

**Fig 3. Univariate predicted probability plots for the outcome of 30 day mortality across the input range of 4 different continuous variables.** This plot demonstrates that the UAN is able to learn the appropriate non-linear risk for each variable in the univariate case, with the risk decreasing with increasing eGFR, but increasing with age, preoperative creatinine and BMI.

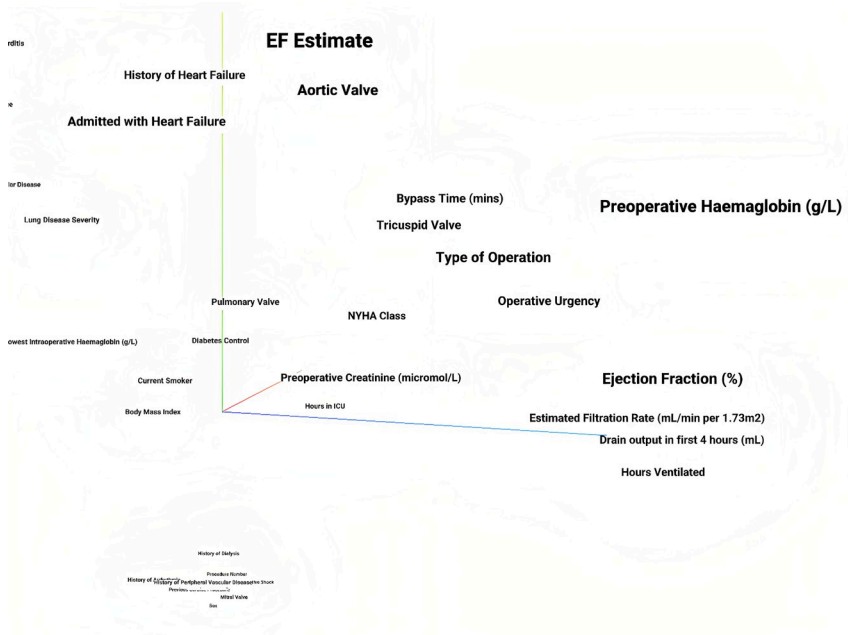

**Fig 4. A visualization in 3 dimensions using t-SNE clustering on the 16 dimensional feature vector representations the UAN learned.** Related variables tended to cluster in 3 dimensional space, for example, history of heart failure clustered with being admitted with heart failure, and type of operation, operative urgency, and bypass time were clustered together.

profiles, which is the first deep-learning model to achieve this outside of using explanatory modeling [29].

Interestingly, trust in the UAN can be furthered by visualizing certain unique characteristics of the UAN. Plotting the univariate probability distribution across the input range of a feature (Fig 3) allows the clinician to know how the model maps a value to a predicted probability. For each feature we interrogated with this strategy, the model learned a mapping matching the expected trend and provided unique insight into the data. For example, the model learned a non-linear distribution for age with a small peak at younger ages, a nadir at 40 years old, and gradual increase accelerating at 70 years old. This pattern correlates with the fact that younger patients requiring cardiac surgery are generally suffering from trauma or congenital disease that confers higher mortality than middle-aged patients undergoing coronary bypass or valve surgery [30, 31]. This is in contrast to a logistic regression model, where the coefficient associated with the feature is limited to a linear relationship [11].

Unique to the UAN model architecture is a feature-specific embedding that maps the feature value to a feature-specific space. By using t-SNE on these embeddings, it is possible to view the clustered relationships between the features (Fig 4). Features that are similar are mapped to a similar space and are therefore clustered. A clinician is able to use this plot to ensure that the UAN learns to cluster features that have similar effects on the outcome. Using both the clustering and univariate probability plots, a clinician can have increased confidence that the model has learned appropriate relationships between the features and the outcome.

## Computational efficiency

A further added benefit to the UAN model is that it is able to model multiple outcomes at the same time. In this study, the model learned the relationship between the inputs and eight

different clinically significant outcomes (see S4 of the supplemental materials). This is in contrast to traditional models, such as LR, where a separate model needs to be trained for each outcome. At inference, the model will generate predictions for all outcomes at the same time as well. The advantage here is the computational cost for training on large datasets for multiple outcomes.

## Limitations

The primary limitations of this study arose from the derivation and external validation datasets. Firstly, clinical variables available for input into a predictive model can be categorized as a modifiable or non-modifiable variable. For example, the variable of age is non-modifiable, whereas preoperative hemoglobin is modifiable by giving red blood cells. Ideally, a predictive model would contain a large number of modifiable risk factors, which would allow the clinician to better optimize patient care pre- or early post-operatively. The ANZSCTS Database was primarily created for auditing surgical outcomes and does not contain a large number of modifiable variables [13]. Future datasets should strive to incorporate more modifiable preoperative variables in order to develop models with more actionable insights. Secondly, there were many differences with how the two databases defined variables. For example, the ANZSCTS Database is updated stochastically with information from clinicians who are directly responsible for patient care, whereas much of the data in the MIMIC III database is derived from codes related to hospital billing [13, 14]. This explains the difference in comorbidity rates for some key variables, such as smoking (Table 1). While this may be a reason for reduced performance on the MIMIC III dataset, these differences may in fact improve the robustness of the external validation results by factoring in differences that healthcare networks across nations may have for defining these variables.

There were also some limitations inherent to the UAN model. The attention function approximates a mixture model, where the output is a mixture of the vector representations of the input variables [16]. This is a modeling assumption that may not accurately reflect the best way to combine the input values.

Finally, although a PN approach to modeling epistemic uncertainty was more effective than GVI, epistemic uncertainty was still poorly modeled compared to aleatoric uncertainty. Future research will be needed to develop and deploy better methods of modeling epistemic uncertainty.

## Conclusion

The UAN is a novel, interpretable, and readily available tool for clinicians to assist in risk stratification in cardiac surgery. The model outperforms current gold standards in important performance benchmarks. Further research needs to be conducted in improving uncertainty calibration and externally validating the model in new cohorts.

## Supporting information

**S1 Table. Variable definitions.** A description of how each variable was defined. (DOCX)

**S2 Table. Outcome definitions.** A description of how each outcome was defined. (DOCX)

**S3 Table. Patient characteristics of the database subsets.** A complete table of patient characteristics for all variables included in the analysis. (DOCX)

**S4 Table. Model performance across all datasets and outcomes.** A complete table for all internal validation and external validation results for performance.
(DOCX)

**S5 Table. Uncertainty calibration across all datasets and outcomes.** A complete table for all internal validation and external validation results for uncertainty calibration.
(DOCX)

**S6 Table. Pairwise T-test p-values for AUC.** Statistical testing for performance differences across cross-validation.
(DOCX)

**S7 Table. Pairwise T-test p-values for sensitivity.** Statistical testing for performance differences across cross-validation.
(DOCX)

**S8 Table. Pairwise T-test p-values for specificity.** Statistical testing for performance differences across cross-validation.
(DOCX)

**S9 Table. Pairwise T-test p-values for AUC of high confidence samples.** Statistical testing for performance differences across cross-validation.
(DOCX)

**S10 Table. Pairwise T-test p-values for AUC of aleatoric uncertainty.** Statistical testing for performance differences across cross-validation.
(DOCX)

**S11 Table. Pairwise T-test p-values for AUC of epistemic uncertainty.** Statistical testing for performance differences across cross-validation.
(DOCX)

**S12 Table. Pairwise T-test p-values for the Brier score.** Statistical testing for performance differences across cross-validation.
(DOCX)

**S1 Appendix. Uncertainty-aware attention network implementation details.** A detailed description of the uncertainty aware network implementation.
(DOCX)

**S1 File.**
(PDF)

**S2 File.**
(PDF)

## Acknowledgments

The ANZSCTS Cardiac Surgery Database Program thanks all of the investigators, data managers, and institutions that participate in the Program.

## Author Contributions

**Conceptualization:** Jahan C. Penny-Dimri, Christoph Bergmeir.

**Data curation:** Christopher M. Reid, Jenni Williams-Spence.

**Formal analysis:** Jahan C. Penny-Dimri.

**Investigation:** Jahan C. Penny-Dimri.

**Methodology:** Jahan C. Penny-Dimri.

**Supervision:** Christoph Bergmeir, Andrew D. Cochrane, Julian A. Smith.

**Writing – original draft:** Jahan C. Penny-Dimri.

**Writing – review & editing:** Jahan C. Penny-Dimri,
Christoph Bergmeir, Christopher M. Reid, Jenni Williams-Spence, Andrew D. Cochrane,
Julian A. Smith.

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
