## [Decision Letter · Decision Letter 0]

27 Aug 2022

PONE-D-22-12461Paying attention to cardiac surgical risk: An interpretable machine learning approach using an uncertainty-aware attentive neural network.PLOS ONE

Dear Dr. Penny-Dimri,

Thank you for submitting your manuscript to PLOS ONE. After careful consideration, we feel that it has merit but does not fully meet PLOS ONE’s publication criteria as it currently stands. Therefore, we invite you to submit a revised version of the manuscript that addresses the points raised during the review process.

We look forward to receiving your revised manuscript.

Kind regards,

Guangyu Tong

Academic Editor

PLOS ONE

Journal Requirements:

2. Will the web application be publicly available upon publication? Please note that PLOS ONE has specific guidelines on code sharing for submissions in which author-generated code underpins the findings in the manuscript. In these cases, all author-generated code must be made available without restrictions upon publication of the work. Please review our guidelines at https://journals.plos.org/plosone/s/materials-and-software-sharing#loc-sharing-code and ensure that your code is shared in a way that follows best practice and facilitates reproducibility and reuse."

3. "Please amend your current ethics statement to address the following concerns:

a) Did participants provide their written or verbal informed consent to participate in this study?

"The ANZSCTS Cardiac Surgery Database Program is funded by the Department of Health (Victoria), the Clinical Excellence Commission (NSW), Queensland Health (QLD), and funding from individual cardiac surgical units participating in the registry. ANZSCTS Database Research activities are supported through a National Health and Medical Research Council Principal Research Fellowship (APP 1136372) and Program Grant (APP 1092642) awarded to C.M. Reid. The Database thanks all of the investigators, data managers, and institutions that participate in the Program."

Additional Editor Comments (if provided):

Two field experts were invited to review your manuscript. The two major areas for improvement are the unclear description of the methods and the unclear significance/contribution of the paper to the existing knowledge. The writing also needs to be significantly improved in order to meet the publication standard.

Reviewers' comments:

Reviewer's Responses to Questions

**Comments to the Author**

1. Is the manuscript technically sound, and do the data support the conclusions?

Reviewer #1: Partly

Reviewer #2: Partly

2. Has the statistical analysis been performed appropriately and rigorously? 

Reviewer #1: I Don't Know

Reviewer #2: I Don't Know

3. Have the authors made all data underlying the findings in their manuscript fully available?

Reviewer #1: No

Reviewer #2: No

4. Is the manuscript presented in an intelligible fashion and written in standard English?

Reviewer #1: No

Reviewer #2: Yes

5. Review Comments to the Author

Reviewer #1: • There are many redundant sentences in the abstract, and the logic is also not clear enough. Please refine the

content of the abstract again. Author need to make clear the research significance of the article, the research

process, the experimental results and conclusions.

• The keywords need to be extracted again, and it could be best to reflect the characteristics of this article.

• The introduction of the article is complex and tedious about the modern approaches, so it is necessary to highlight

the research status at home and abroad, and simplify the background of the article.

• In this manuscript, the explanation of flow diagram in Figure 1 is not enough, and the author needs to give a more

detailed explanation to facilitate the understanding of readers.

• The analysis of the experimental data in Table 3 can be added to improve the uncertainty calibration calculation.

• The conclusion of this manuscript needs to be reorganized by highlighting the research results of this research

work, not too much introduction of future research methods.

• There are grammatical errors in this article, which need to be corrected after careful checking.

• Carefully check the references of the article and also ensure the citation is in the right place.

• The objectives of your work need to be clearly stated in different sections.

• Author needs to highlight the novelty of this research work in clear manner.

Reviewer #2: The given manuscript by Penny-Dimri et al. highlights a relevant subject and presents a potentially beneficial and useful method. The results seem convincing. However, some issues need to be addressed before the manuscript can be published.

Major issues (in decreasing order of importance):

- The UAN approach should be described in detail, as this is not a commonly used method. Please provide details about the architecture, its internals and the training procedure.

- In Tables 2 and 3, the best results are written in bold. Does that mean that they are simply the best or does that mean that the bold results are significantly better than other results? I strongly suggest to test for significance and to describe the employed statistical procedure in detail.

- It is imperative (otherwise a bias is introduced that renders the results invalid) that the calibration/training of missing value imputation methods is only done on the training set(s). Was that the case in the third strategy mentioned in the manuscript? It did not become entirely clear. Please provide more detail!

- Include more recent reference to works dealing with machine learning in cardiac surgery, e.g.:

Bodenhofer et al., Eur. J. Cardiothorac. Surg. 60(6), 2021. https://doi.org/10.1093/ejcts/ezab219

Jiang et al., Front. Cardiovasc. Med. 8, 2021. https://doi.org/10.3389/fcvm.2021.771246

Shuhaiber & Conte, Eur. J. Cardiothorac. Surg. 60(6), 2021. https://doi.org/10.1093/ejcts/ezab324

Some of these works indeed demonstrate a general advantage of more complex methods over LR.

- There are also some deeper works dealing with interpretations of machine learning models (some limited to tree ensembles, some model-agnostic), e.g.:

Lundberg & Lee, NIPS 2017. https://papers.nips.cc/paper/2017/file/8a20a8621978632d76c43dfd28b67767-Paper.pdf

Sharma et al., arXiv:2012.06734, 2020. https://arxiv.org/pdf/2010.06734.pdf

Minor issues:

- The figures have very poor resolutions. In particular, the heatmaps in Fig. 2 are illegible. Please fix this!

6. PLOS authors have the option to publish the peer review history of their article (what does this mean?). If published, this will include your full peer review and any attached files.

Reviewer #1: No

Reviewer #2: No

---

## [Author Response · Author response to Decision Letter 0]

4 Oct 2022

EDITORIAL COMMENTS:

Response:

Thank you for the above resources. The manuscript has been formatted accordingly.

2. Will the web application be publicly available upon publication? Please note that PLOS ONE has specific guidelines on code sharing for submissions in which author-generated code underpins the findings in the manuscript. In these cases, all author-generated code must be made available without restrictions upon publication of the work. Please review our guidelines at https://journals.plos.org/plosone/s/materials-and-software-sharing#loc-sharing-code and ensure that your code is shared in a way that follows best practice and facilitates reproducibility and reuse."

Response:

The code base for generating the web application and running the experiments will be available as a public GitHub repository (https://github.com/jahanpd/UAN).

3. "Please amend your current ethics statement to address the following concerns:

a) Did participants provide their written or verbal informed consent to participate in this study?

Response:

The ethics statement has been amended as requested to address the question of consent. The ANZSCTS National Database program data policy and patient information sheet have been included as supplementary materials. See lines 210-213.

"The ANZSCTS Cardiac Surgery Database Program is funded by the Department of Health (Victoria), the Clinical Excellence Commission (NSW), Queensland Health (QLD), and funding from individual cardiac surgical units participating in the registry. ANZSCTS Database Research activities are supported through a National Health and Medical Research Council Principal Research Fellowship (APP 1136372) and Program Grant (APP 1092642) awarded to C.M. Reid. The Database thanks all of the investigators, data managers, and institutions that participate in the Program."

Response:

Thank you for highlighting this apparent discrepancy. It should be noted that the database operates under a grant structure with its own ethics approval, whereas the research of this paper is separate from the operations of the database organization. This work received no specific funding. We have addressed this separation of concerns in our funding statement and removed the funding statement from the acknowledgments.

REVIEWER COMMENTS (Tasks Highlighted):

5. Review Comments to the Author

Reviewer #1: 

• There are many redundant sentences in the abstract, and the logic is also not clear enough. Please refine the content of the abstract again. Author need to make clear the research significance of the article, the research process, the experimental results and conclusions.

Response:

Thank you for your feedback. The abstract has been rewritten to address these concerns and fulfill PLOS One formatting requirements.

• The keywords need to be extracted again, and it could be best to reflect the characteristics of this article.

Response:

Thank you for your comments. We have revised our keywords.

• The introduction of the article is complex and tedious about the modern approaches, so it is necessary to highlight the research status at home and abroad, and simplify the background of the article.

Response:

Thank you for your feedback. We have amended our introduction to simplify the discussion of alternative approaches and moved some of the method details to the methods section.

• In this manuscript, the explanation of flow diagram in Figure 1 is not enough, and the author needs to give a more detailed explanation to facilitate the understanding of readers.

Response:

Thank you for your feedback. In-text reference to Figure 1 has been made on line 140 in order to provide in-text context to the visualized data processing pipeline in Figure 1.

• The analysis of the experimental data in Table 3 can be added to improve the uncertainty calibration calculation.

Response:

Thank you for your feedback. We have added references to Table 3 at lines 232 and 236.

• The conclusion of this manuscript needs to be reorganized by highlighting the research results of this research work, not too much introduction of future research methods.

Response:

Thank you for your feedback. The discussion and conclusion have been edited and reorganized to focus more on the research presented.

• There are grammatical errors in this article, which need to be corrected after careful checking.

Response:

Thank you for your feedback. We have parsed our work and corrected our grammar.

• Carefully check the references of the article and also ensure the citation is in the right place.

Response:

Thank you for your feedback. We have re-evaluated the references list.

• The objectives of your work need to be clearly stated in different sections.

Response:

Thank you for your feedback. Aims and hypotheses have been altered to be more clear.

• Author needs to highlight the novelty of this research work in clear manner.

Response:

Thank you for your feedback. The novelty of this work has been highlighted on lines 267, 281, 341. Additionally, we hope that the further supplementary materials provided to explain the model in greater detail aid the readers in appreciating the novelty of our approach.

Reviewer #2: The given manuscript by Penny-Dimri et al. highlights a relevant subject and presents a potentially beneficial and useful method. The results seem convincing. However, some issues need to be addressed before the manuscript can be published.

Major issues (in decreasing order of importance):

- The UAN approach should be described in detail, as this is not a commonly used method. Please provide details about the architecture, its internals, and the training procedure.

Response:

Thank you for your feedback. We have added further details regarding the model design and the training algorithm in the supplemental materials (S1 Appendix) with in-text reference at line 162.

- In Tables 2 and 3, the best results are written in bold. Does that mean that they are simply the best or does that mean that the bold results are significantly better than other results? I strongly suggest testing for significance and describing the employed statistical procedure in detail.

Response:

Thank you for your suggestion. The bold results are simply the best performing. Regarding the application of a statistical test on cross-validation, this is a relatively controversial topic. While cross-validation and other bootstrap methods are widely used to assess the generalization performance of machine learning and statistical models, the process is well known to not produce independent samples. This arises as each fold will share data across training and test sets. As such, applying traditional statistical tests, such as a student's T-test will produce excess type 1 error. While there are corrected tests and cross-validation schemes that purport to improve the power of these tests, we suggest that these come burdened with their own assumptions and do not solve the underlying problem (some references that highlight the problems and approaches include: https://doi.org/10.48550/arXiv.2104.00673, https://doi.org/10.1023/A:1024068626366, https://doi.org/10.1162/neco.1997.9.6.1245, https://doi.org/10.1162/089976698300017197
https://doi.org/10.1007/978-3-540-24775-3_3). Applying a statistical test may give an unreliable p-value and we fear reporting such results would give the reader a false assurance. We also note that it is quite common in ML literature to simply report the raw performance. We have chosen to report the mean of the cross-validation with an accompanying measure of the variance. We believe this gives readers enough information to assess the distribution of performance for each model.

- It is imperative (otherwise a bias is introduced that renders the results invalid) that the calibration/training of missing value imputation methods is only done on the training set(s). Was that the case in the third strategy mentioned in the manuscript? It did not become entirely clear. Please provide more detail!

Response:

Thank you for your feedback. Indeed missing value imputation was performed only on the training set data during cross-validation. This has been specified in-text at line 181, and is verifiable from the provided codebase.

- Include more recent references to works dealing with machine learning in cardiac surgery, e.g.:

Response:

Thank you for your suggestions. We have cited two recent meta-analyses, the latter which includes Bodenhofer et al. We have added the additional citations as requested, however our phrasing remains conservative as the most recent meta-analysis on this topic failed to find a significant advantage of ML over LR (https://doi.org/10.1111/jocs.16842). Lundberg and Lee and Sharme and Conte have been cited where we reference explainability on line 279.

Minor issues:

- The figures have very poor resolutions. In particular, the heatmaps in Fig. 2 are illegible. Please fix this!

Response:

Thank you for your feedback. We have endeavored to reformat the images as per PLOS One guidelines and using the PACE online tool to ensure adequate formatting. Please note the appearance of the figures in the compiled PDF do not reflect the quality of the tiff files when downloaded separately.

---

## [Decision Letter · Decision Letter 1]

28 Nov 2022

PONE-D-22-12461R1Paying attention to cardiac surgical risk: An interpretable machine learning approach using an uncertainty-aware attentive neural network.PLOS ONE

Dear Dr. Penny-Dimri,

Thank you for submitting your manuscript to PLOS ONE. After careful consideration, we feel that it has merit but does not fully meet PLOS ONE’s publication criteria as it currently stands. Therefore, we invite you to submit a revised version of the manuscript that addresses the points raised during the review process.

We look forward to receiving your revised manuscript.

Kind regards,

Guangyu Tong

Academic Editor

PLOS ONE

Additional Editor Comments:

Please carefully address the comments from Reviewer 2 and ensure the claim of this paper is not overstated and biased, as pointed out by Reviewer 2. I also recommend the authors perform the additional test Reviewer 2 requested.

Reviewers' comments:

Reviewer's Responses to Questions

**Comments to the Author**

1. If the authors have adequately addressed your comments raised in a previous round of review and you feel that this manuscript is now acceptable for publication, you may indicate that here to bypass the “Comments to the Author” section, enter your conflict of interest statement in the “Confidential to Editor” section, and submit your "Accept" recommendation.

Reviewer #2: (No Response)

2. Is the manuscript technically sound, and do the data support the conclusions?

Reviewer #2: Yes

3. Has the statistical analysis been performed appropriately and rigorously? 

Reviewer #2: I Don't Know

4. Have the authors made all data underlying the findings in their manuscript fully available?

Reviewer #2: Yes

5. Is the manuscript presented in an intelligible fashion and written in standard English?

Reviewer #2: Yes

6. Review Comments to the Author

Reviewer #2: Most of my comments have properly been taken into account by the authors, so I can recommend the manuscript for publication now.

There are two issues, however, that I still want to comment on:

1) I agree that significance of cross validation results are a subtle topic. I acknowledge that p-values might be inflated, so the tests might have a higher type I error rate. It is a fact, though, that such tests are still quite common in literature. Let me make this point clear: I was not so much afraid of insignificant differences being falsely flagged as significant. My concern would rather have been the following: If a result marked in bold would fail the test for significance, then it has no justification to be reported as a serious advancement. So I was rather afraid of falsely announcing an algorithm as better than the rest although it is actually on par with another. Apart from that, I have no doubt - looking at the numbers and the standard deviations in parentheses - that the significance test would be successful. So I suggest the authors either consider performing the test (despite its weaknesses and limitations) or provide a short argumentation in the manuscript why they have not done it (even though it is quite commeon).

2) Regarding the citations on machine learning in cardiac surgery that I suggested: I am not in the position to force the authors to cite particular papers, so I accept that they have not followed my suggestion. However, since they mention their own meta study (which has very much attracted my interest when it appeared), I want to take this opportunity to comment on this work in relation to the present manuscript under review. First of all, I pretty much disagree with their earlier meta-study. I think it deals with an ill-posed question. Nobody every claimed that ML gives a general edge over LR. I fully agree that there might be some situations in which LR is sufficient or even advisable (less overfitting, better interpretability). However, there might also be situations in which ML indeed is advantageous, in particular in situations with complex, non-linear interactions between the input features. I try not to be too polemic about this issue, but the meta-study appears to me like asking whether there is evidence that trucks are better than cars. Well, in many cases, no. But there are some case where the additional power and loading capacity of a truck is of great advantage. It depends on the situation. It is now really ironic/funny/whatever that the authors of exactly this meta-study try to get a paper about ML in cardical surgery published. No matter whether you cite your own meta-study, I think this asks for an explanation.

7. PLOS authors have the option to publish the peer review history of their article (what does this mean?). If published, this will include your full peer review and any attached files.

Reviewer #2: No

---

## [Author Response · Author response to Decision Letter 1]

27 May 2023

Additional Editor Comments:

Please carefully address the comments from Reviewer 2 and ensure the claim of this paper is not overstated and biased, as pointed out by Reviewer 2. I also recommend the authors perform the additional test Reviewer 2 requested.

Response:

Thank-you for your feedback. Statistical testing has been performed and is available in the supplemental materials (see response to reviewer 2).

Reviewers' comments:

Reviewer #2: Most of my comments have properly been taken into account by the authors, so I can recommend the manuscript for publication now.

There are two issues, however, that I still want to comment on:

1) I agree that significance of cross validation results are a subtle topic. I acknowledge that p-values might be inflated, so the tests might have a higher type I error rate. It is a fact, though, that such tests are still quite common in literature. Let me make this point clear: I was not so much afraid of insignificant differences being falsely flagged as significant. My concern would rather have been the following: If a result marked in bold would fail the test for significance, then it has no justification to be reported as a serious advancement. So I was rather afraid of falsely announcing an algorithm as better than the rest although it is actually on par with another. Apart from that, I have no doubt - looking at the numbers and the standard deviations in parentheses - that the significance test would be successful. So I suggest the authors either consider performing the test (despite its weaknesses and limitations) or provide a short argumentation in the manuscript why they have not done it (even though it is quite commeon).

Thank-you for your ongoing review. We have provided statistical testing using T-tests and amended the manuscript to reflect this at lines 182 and 240. Tables of p-values for the pairwise T-tests are provided in the supplementary materials.

2) Regarding the citations on machine learning in cardiac surgery that I suggested: I am not in the position to force the authors to cite particular papers, so I accept that they have not followed my suggestion. However, since they mention their own meta-study (which has very much attracted my interest when it appeared), I want to take this opportunity to comment on this work in relation to the present manuscript under review. First of all, I pretty much disagree with their earlier meta-study. I think it deals with an ill-posed question. Nobody every claimed that ML gives a general edge over LR. I fully agree that there might be some situations in which LR is sufficient or even advisable (less overfitting, better interpretability). However, there might also be situations in which ML indeed is advantageous, in particular in situations with complex, non-linear interactions between the input features. I try not to be too polemic about this issue, but the meta-study appears to me like asking whether there is evidence that trucks are better than cars. Well, in many cases, no. But there are some case where the additional power and loading capacity of a truck is of great advantage. It depends on the situation. It is now really ironic/funny/whatever that the authors of exactly this meta-study try to get a paper about ML in cardical surgery published. No matter whether you cite your own meta-study, I think this asks for an explanation.

Response:

We thank the reviewer for their feedback. Additionally, we appreciate their analysis of our previous work and wholeheartedly agree with their conclusion. In particular, we wish to draw their attention to the discussion of the meta-analysis where we make precisely the same point. This is to say that although ML does not outperform LR as defined by the C-index, there are many reasons to consider using an ML model in non-linear settings such as peri-operative risk prediction. We suggest that ML can add value to risk prediction beyond raw performance. We make explicit the advantages of using explanatory modelling, such as with Shapley values. Furthermore, we believe that our submission under review adds to this argument by showing the flexibility to missing data, and unique explanatory visualization, which our novel ML approach provides.

---

## [Decision Letter · Decision Letter 2]

31 Jul 2023

Paying attention to cardiac surgical risk: An interpretable machine learning approach using an uncertainty-aware attentive neural network.

PONE-D-22-12461R2

Dear Dr. Penny-Dimri,

We’re pleased to inform you that your manuscript has been judged scientifically suitable for publication and will be formally accepted for publication once it meets all outstanding technical requirements.

Kind regards,

Guangyu Tong

Academic Editor

PLOS ONE

Additional Editor Comments (optional):

Reviewers' comments:

Reviewer's Responses to Questions

**Comments to the Author**

1. If the authors have adequately addressed your comments raised in a previous round of review and you feel that this manuscript is now acceptable for publication, you may indicate that here to bypass the “Comments to the Author” section, enter your conflict of interest statement in the “Confidential to Editor” section, and submit your "Accept" recommendation.

Reviewer #2: All comments have been addressed

2. Is the manuscript technically sound, and do the data support the conclusions?

Reviewer #2: Yes

3. Has the statistical analysis been performed appropriately and rigorously? 

Reviewer #2: Yes

4. Have the authors made all data underlying the findings in their manuscript fully available?

Reviewer #2: Yes

5. Is the manuscript presented in an intelligible fashion and written in standard English?

Reviewer #2: Yes

6. Review Comments to the Author

Reviewer #2: The authors have taken into account my comments properly (or have responded adequately). So I recommend this manuscript for publication.

7. PLOS authors have the option to publish the peer review history of their article (what does this mean?). If published, this will include your full peer review and any attached files.

Reviewer #2: No

---

## [Editor Report · Acceptance letter]

10 Aug 2023

PONE-D-22-12461R2 

Paying attention to cardiac surgical risk: An interpretable machine learning approach using an uncertainty-aware attentive neural network. 

Dear Dr. Penny-Dimri:

I'm pleased to inform you that your manuscript has been deemed suitable for publication in PLOS ONE. Congratulations! Your manuscript is now with our production department. 

Kind regards, 

on behalf of

Dr. Guangyu Tong 

Academic Editor

PLOS ONE